# Development of Lightweight and High-Performance Ballistic Helmet Based on Poly(Benzoxazine-co-Urethane) Matrix Reinforced with Aramid Fabric and Multi-Walled Carbon Nanotubes

**DOI:** 10.3390/polym12122897

**Published:** 2020-12-03

**Authors:** Jusmin Daungkumsawat, Manunya Okhawilai, Krittapas Charoensuk, Radhitya Banuaji Prastowo, Chanchira Jubsilp, Panagiotis Karagiannidis, Sarawut Rimdusit

**Affiliations:** 1Polymer Engineering Laboratory, Department of Chemical Engineering, Faculty of Engineering, Chulalongkorn University, 254 Phayathai Road, Pathumwan, Bangkok 10330, Thailand; jusmin_dksw@outlook.com (J.D.); immtbcon@gmail.com (K.C.); radhityabprastowo@gmail.com (R.B.P.); 2Metallurgy and Materials Science Research Institute, Chulalongkorn University, Bangkok 10330, Thailand; Manunya.O@chula.ac.th; 3Research Unit on Polymeric Materials for Medical Practice Devices, Chulalongkorn University, Bangkok 10330, Thailand; 4Department of Chemical Engineering, Faculty of Engineering, Srinakharinwirot University, Nakhonnayok 26120, Thailand; chanchira@g.swu.ac.th; 5School of Engineering, Faculty of Technology, University of Sunderland, Sunderland SR6 0DD, UK; panagiotis.karagiannidis@sunderland.ac.uk

**Keywords:** poly(benzoxazine-co-urethane), MWCNTs, ballistic helmet, simulation

## Abstract

This study aims to develop a lightweight ballistic helmet based on nanocomposite with matrix of the copolymer of benzoxazine with an urethane prepolymer [poly(BA-a-co-PU)], at mass ratio 80/20, reinforced with aramid fabric and multi-walled carbon nanotubes (MWCNTs). This has a protection level II according to the National Institute of Justice (NIJ) 0106.01 standard. The effects of MWCNTs mass content in a range of 0 to 2 wt% on tensile, physical and ballistic impact properties of the nanocomposite were investigated. The results revealed that the introduction of MWCNTs enhanced the tensile strength and energy at break of the nanocomposite; the highest values were obtained at 0.25 wt%. In addition, the nanocomposite laminate with 20 plies of aramid fabric showed the lowest back face deformation of 8 mm which was much lower than that specified by the NIJ standard. According to Military Standard (MIL-STD) 662F, the simulation prediction revealed that the ballistic limit of the ballistic helmet nanocomposite was as high as 632 m s^−1^. The developed laminates made of aramid fabric impregnated with poly(BA-a-co-PU) 80/20 containing 0.25 wt% MWCNTs showed great promise for use as a light weight and high-performance ballistic helmet.

## 1. Introduction

Humans develop threatening weapons such as guns, rifles, explosive powder and other highly destruction weapon systems. Therefore, the personal protective equipment, especially in the case of the ballistic helmet, is designed to meet high standards to ensure maximum protection and minimize the mortality of people. Even though the head and neck sections only comprise around 12% of the body area, they encounter up to 25% of all reported projectile hits [1]. Traumatic brain injury can occur not only by penetration, but also by the energy transferred to the helmet due to the retarded projectile through the interior foams, causing back face deformation (BFD) [1,2]. An effective helmet should not only be able to stop penetration, but also must minimize injuries from BFD and be lightweight. In the past, helmets were made of steel to protect against bullets and blast impacts. However, their heavy weight is a crucial drawback. Nowadays, helmets based on polymer composites have been developed to substitute steel with increased resistance capabilities and lower weight which make the soldier more effective fighters with higher mobility [3].

High-performance fibers made of ultra-high molecular weight polyethylene (UHMWPE) or aramid fibers have been widely used to weave ballistic-resistant fabrics. Although UHMWPE fibers are better than aramid fibers, as far as the absorbed energy under high velocity impacts (900 m s^−1^) and have lower weight, the UHMWPE fabrics exhibit lower creep resistance and are more expensive than aramid fabrics [4]. Aramid fabrics exhibit outstanding properties, such as low density, high tensile properties, good heat and flame resistance. Thermoset polymers such as epoxy [5], polyester [6] and phenolics [7] are usually used as resin matrices for ballistic impact applications. Benzoxazine resin (BA-a) is a newly developed phenolic resin from the reaction of bisphenol-A, paraformaldehyde and aniline, without the use of solvent and catalyst. BA-a has low viscosity and near-zero volume shrinkage during curing, and the prepared poly(BA-a) shows excellent mechanical and thermal properties useful for a ballistic matrix. Moreover, BA-a can be copolymerized with various types of resins to meet application requirements. Rimdusit et al. reported that BA-a copolymerized with urethane prepolymer (PU) to make poly(benzoxazine-co-urethane) or poly(BA-a-co-PU), which demonstrated synergistic behavior in flexural and tensile strength [8,9]. Moreover, Okhawilai et al. investigated the ballistic performance of poly(BA-a-co-PU) matrix reinforced with aramid fabric as a soft armor application and found that 20 wt% PU provided the greatest energy absorption, indicating the greatest ballistic impact protection [4,8].

Carbon-based nanomaterials including fullerenes, graphene and related materials and carbon nanotubes (CNTs) [10] are among the most used reinforcing agents. Due to the remarkable properties of CNTs, such as optical, electrical, thermal, chemical and especially mechanical, CNT-reinforced nanocomposites have been widely developed [11,12]. CNTs have not only extremely high modulus and stiffness, but also low density, hence they are the first and most ideal choice among reinforcing agents, where high mechanical and very lightweight properties are required [13,14]. The high energy absorption capability of CNTs makes them attractive materials for ballistic applications. Lauranzi et al. investigated the absorbed energy during ballistic impact of epoxy-Kevlar 29 panel reinforced with 0.5 wt% multi-walled carbon nanotubes (MWCNTs). They found that the ballistic impact behavior of this composite was enhanced due to the high energy absorption and energy dissipation of MWCNTs [15,16].

In a recent review [17], it is reported that the modern design process of simulation methods is widely used for a reduction in time-consuming and costly experimental work. Computer modelling is convenient, reliable and can provide much more information in research of ballistic applications [4,18]. Rodriguez-Millan et al. [2] developed a simulation finite element of ballistic impact on plates and on ballistic helmet. The experimental work was carried out to determine the material properties and validate the ballistic performance of the real helmet. Simulation results showed the accuracy of the model and its suitability for use as a design tool [2].

Therefore, the objective of this research is to develop a lightweight and high-performance ballistic helmet from nanocomposites. The nanocomposite specimens fabricated from the poly(BA-a-co-PU) matrix having 20 wt% of PU reinforced with aramid fabric and different MWCNT content ranging from 0 to 2 wt% were mechanically and physically characterized. The introduction of MWCNTs is expected to absorb and dissipate impact energy from the projectile effectively, thus resulting in no perforation with low BFD on the ballistic helmet. The obtained nanocomposite laminates with various numbers of plies were then evaluated in terms of their ballistic performance following the NIJ standard 0106.01 level II. The failure behavior of the specimens was studied by performing computational modeling using ANSYS AUTODYN. The BFD on four impact locations, i.e., top, front, back, and side of the ballistic helmet was predicted according to NIJ0106.01 standard level II, as well as the ballistic limit (V_50_) following Military Standard (MIL-STD) 662F.

## 2. Materials and Methods

### 2.1. Materials

BA-a is based on bisphenol A, aniline and para-formaldehyde. Bisphenol A (polycarbonate grade) was provided by PTT Phenol Co., Ltd (Rayong, Thailand). Aniline (AR grade), and para-formaldehyde (AR grade) were purchased from Loba Chemical PVT. LTD. (Mumbai, India) and Merck Company (Kenilworth, NJ, USA), respectively. Toluene diisocyanate (TDI) and polypropylene polyol with a molecular weight of 2000 g mol^−1^ were provided by IRPC Public Company Limited. Aramid fabric with an areal density of 340 g m^−2^ was supplied by Thai Polyadd Limited Partnership (Bangkok, Thailand). MWCNTs, with an outer diameter of 12.9 nm and length of 3–12 µm, were purchased from Nano Generation Company Limited, Bangkok, Thailand.

### 2.2. Nanocomposite Fabrication

#### 2.2.1. Preparation of BA-A and PU

BA-a resin was synthesized from bisphenol-A, para-formaldehyde and aniline at a molar ratio of 1:4:2 at a temperature of 110 °C for 40 min following the patented solventless technology [19]. The synthesis of BA-a resin is shown in Scheme 1 [8]. PU was prepared by mixing TDI and polypropylene glycol with an average molecular weight of 2000 g mol^−1^ in a four-necked round-bottomed flask under a nitrogen stream at a temperature of 70 °C for 40 min to yield a light yellow prepolymer.

#### 2.2.2. Preparation of Nanocomposite Laminates

Nanocomposites were prepared by adding MWCNTs at desired mass content (0–2 wt%) into the BA-a/PU compound at a mass ratio of 80/20 at a temperature of 110 °C. The prepared molding compound was subsequently coated on aramid fabrics. The mass content of aramid fabrics in the nanocomposites was approximately 80 wt%. The nanocomposite laminates were obtained using compression molding at a temperature of 200 °C and a pressure of 100 bar for 2 h. During this step polymerization reaction between BA-a monomer and PU, prepolymer takes place to give poly(BA-a-co-PU) [4,8]. The poly(BA-a-co-PU) structure is shown in Scheme 2 [8].

### 2.3. Characterization

Tensile tests on 8-ply nanocomposite laminates were carried out based on ASTM D3039. Specimens with dimensions of 150 × 25 × 3.15 mm^3^ were prepared. Measurements were performed with a cross head speed of 2 mm min^−1^ using a universal testing machine model 8872, Instron Co., Ltd., Bangkok, Thailand.

The morphological properties of the nanocomposites were evaluated by scanning electron microscopy (SEM) at 15 kV using a JSM-6510A microscope from JEOL Ltd. (Tokyo, Japan). The test specimens were coated with gold prior to measurements.

Ballistic tests on the nanocomposite laminates were carried out according to NIJ-STD-0106.01 level II standard using a 9 mm full metal jacketed (FMJ) projectile with a nominal bullet mass of 8.0 g with velocities in a range of 394–407 m s^−1^ [20]. The number of plies for ballistic tests were varied between 10 and 20 plies. The dimensions of the specimens were 150 × 150 mm^2^. Each panel was impacted with one shot. The ballistic impact resistance and failure behavior of the nanocomposite laminates were evaluated.

### 2.4. Computational Modeling

The ballistic impact performance of the nanocomposite laminate was evaluated using ANSYS AUTODYN. The properties of nanocomposites followed an orthotropic material model in which the properties along orthogonal planes are different. The orthotropic equation of state (EOS) is presented in Equation (1) [21]. The total stress, $\sigma_{ij}$, is related with the total strain, $\varepsilon_{ij}$, through the stiffness matrix, $C_{ij}$. The stiffness matrix coefficient, *C* is a function of Young’s modulus, $E_{ij}$, shear modulus, $G_{ij}$, and Poisson’s ratio, $\nu_{ij}$, as shown in Equation (2).
(1)$$\left[\begin{array}{c} \sigma_{11}\\ \sigma_{22}\\ \sigma_{33}\\ \sigma_{23}\\ \sigma_{31}\\ \sigma_{12} \end{array}\right]=\left[\begin{array}{cccccc} C_{11} & C_{12} & C_{13} & 0 & 0 & 0\\ C_{21} & C_{22} & C_{23} & 0 & 0 & 0\\ C_{31} & C_{32} & C_{33} & 0 & 0 & 0\\ 0 & 0 & 0 & C_{44} & 0 & 0\\ 0 & 0 & 0 & 0 & C_{55} & 0\\ 0 & 0 & 0 & 0 & 0 & C_{66} \end{array}\right]\left[\begin{array}{c} \varepsilon_{11}\\ \varepsilon_{22}\\ \varepsilon_{33}\\ \varepsilon_{23}\\ \varepsilon_{31}\\ \varepsilon_{12} \end{array}\right],$$
(2)$$[C]=\left[\begin{array}{cccccc} \frac{1}{E_{11}} & \frac{-\nu_{12}}{E_{22}} & \frac{-\nu_{31}}{E_{33}} & 0 & 0 & 0\\ \frac{-\nu_{12}}{E_{22}} & \frac{1}{E_{22}} & \frac{-\nu_{23}}{E_{22}} & 0 & 0 & 0\\ \frac{-\nu_{31}}{E_{33}} & \frac{-\nu_{23}}{E_{22}} & \frac{1}{E_{33}} & 0 & 0 & 0\\ 0 & 0 & 0 & \frac{1}{2G_{12}} & 0 & 0\\ 0 & 0 & 0 & 0 & \frac{1}{2G_{23}} & 0\\ 0 & 0 & 0 & 0 & 0 & \frac{1}{2G_{31}} \end{array}\right],$$

The total strains are split into average strain, $\varepsilon_{avg}$, and deviatoric strain, $\varepsilon_{ij}^{d}$, following Equation (3). The average strain is calculated from a third of the trace of the strain tensor (4).
(3)$$\varepsilon_{ij}=\varepsilon_{avg}+\varepsilon_{ij}^{d},$$
(4)$$\varepsilon_{avg}=\frac{1}{3}(\varepsilon_{11}+\varepsilon_{22}+\varepsilon_{33})=\frac{1}{3}\varepsilon_{vol},$$

The total stress is given by Equations (5)–(7),
(5)$$\sigma_{11}=\frac{1}{3}(C_{11}+C_{12}+C_{13})\varepsilon_{vol}+C_{11}\varepsilon_{11}^{d}+C_{12}\varepsilon_{22}^{d}+C_{13}\varepsilon_{33}^{d},$$
(6)$$\sigma_{22}=\frac{1}{3}(C_{21}+C_{22}+C_{23})\varepsilon_{vol}+C_{21}\varepsilon_{11}^{d}+C_{22}\varepsilon_{22}^{d}+C_{23}\varepsilon_{33}^{d},$$
(7)$$\sigma_{33}=\frac{1}{3}(C_{31}+C_{32}+C_{33})\varepsilon_{vol}+C_{31}\varepsilon_{11}^{d}+C_{32}\varepsilon_{22}^{d}+C_{33}\varepsilon_{33}^{d},$$

The pressure that is defined as the average of total stress in opposite direction can be calculated by Equations (8) and (9); this gives the pressure which is related with deviatoric strain and volumetric strain as presented in (9).
(8)$$P=-\frac{1}{3}(\sigma_{11}+\sigma_{22}+\sigma_{33}),$$
(9)$$P=-\frac{1}{9}[C_{11}+C_{22}+C_{33}+2(C_{12}+C_{23}+C_{31})]\varepsilon_{vol}-\frac{1}{3}[C_{11}+C_{12}+C_{13}]\varepsilon_{11}^{d}$$
$$-\frac{1}{3}[C_{21}+C_{22}+C_{23}]\varepsilon_{22}^{d}-\frac{1}{3}[C_{31}+C_{32}+C_{33}]\varepsilon_{33}^{d},$$

For an isotropic material, the first term of (10) can be replaced with the Mie–Gruneisen (Shock) EOS as shown in Equation (11).
(10)$$P=p_{ref}+\Gamma \rho (e-e_{ref})-\frac{1}{3}[C_{11}+C_{12}+C_{13}]\varepsilon_{11}^{d}-\frac{1}{3}[C_{21}+C_{22}+C_{23}]\varepsilon_{22}^{d}-\frac{1}{3}[C_{31}+C_{32}+C_{33}]\varepsilon_{33}^{d},$$

The 9 mm FMJ bullet used for simulation in this research is shown in Figure 1a. The FMJ bullet is made of a copper jacket and a lead core. The response of these two materials under ballistic impact is explained using the Mie–Gruneisen EOS which gives the relation between pressure, density and specific internal energy [21] defined by Equation (9).
(11)$$p=\frac{\rho_{0}C_{1}^{2}\eta }{{\left(1-\eta S_{1}\right)}^{2}}\left(1-\frac{\eta \Gamma_{0}}{2}\right)+\Gamma_{0}\rho_{0}e,$$
where pressure p depends on density and temperature as described in [21], $\rho_{0}$ is the reference density, $C_{1}$ is the bulk speed sound and η is the nominal volumetric compressive strain which is defined as
(12)$$\eta =\frac{1-\rho_{0}}{\rho },$$
where ρ is the current density, $S_{1}$ is the Gruneisen parameter, $\Gamma_{0}$ is the Gruneisen coefficient and e is an internal energy.

The material properties used in this research were taken from ANSYS AUTODYN library as presented in Table 1 [22].

The geometries and mesh creation of the nanocomposite for ballistic impact performance test are shown in Figure 1b. The simulation work was carried out in the same manner as the experimental work. The deformation area of the nanocomposite laminate observed from both simulation and experimental work was compared to validate the input parameter of the nanocomposite material in which this technique is convenience and widely used for parameter validation with acceptable errors [2,23,24].

The performances of the ballistic helmet to withstand 9 mm FMJ at four sides including top, front, back, and side were predicted according to the NIJ-STD-0106.01. The geometry and shot locations on the ballistic helmet are shown in Figure 2. The EOS and input material properties were the same as in the ballistic nanocomposite laminate. The BFD occurred on the inner side of the helmet, after that, the ballistic impact was then measured.

For prediction of ballistic limit (V_50_) using simulation method according to the MIL-STD-662F standard [25], the ballistic helmet was subjected to fragment simulating projectile (FSP) made of steel AISI4340, with weight of 1.1 g and initial velocity of 610 m s^−1^. The FSP was modeled following the Johnson–Cook model describing large strains, high strain rates and high temperature responsible for metallic materials under ballistic impact. The yield stress definition of Johnson–Cook is given by Equation (13).
(13)$$\sigma_{y}=[A+B{\bar{\varepsilon }}_{p}^{n}][1+C\ln({\dot{\varepsilon }}^{*})][1-{\left(T^{*}\right)}^{m}],$$
where ${\dot{\varepsilon }}^{*}=\frac{{\dot{\bar{\varepsilon }}}_{p}}{{\dot{\varepsilon }}_{0}}$ and $T^{*}=\frac{T-T_{r}}{T_{m}-T_{r}}$. A is yield stress, B is hardening constant, n is the hardening exponent, ${\bar{\varepsilon }}_{p}$ is the effective plastic strain, ${\dot{\bar{\varepsilon }}}_{p}$ is the effective plastic strain rate, ${\dot{\varepsilon }}_{0}$ is the reference strain rate, normally defined as 1.0 s^−1^, C is a strain rate coefficient, m is a thermal softening exponent, $T_{r}$ is room temperature (300 K) and $T_{m}$ is melting temperature (1793 K).

The properties for steel 4340 FSP are summarized in Table 2 which were taken from the AUTODYN library [22]. The geometry of FSP is shown in Figure 3. The ballistic limit is defined as an average of six impact velocities; three impact velocities resulted in partial and three impact velocities resulted in complete penetration.

## 3. Results and Discussion

### 3.1. Tensile Properties of Nanocomposite Laminates

During ballistic impact, the nanocomposite laminate failed by several mechanisms including matrix cracking, delamination, shear plugging and especially tensile failure, which is reported to be the major failure, causing the breakage of the fabric [26]. For this reason, tensile tests of the prepared nanocomposite laminates were carried out. The results of tensile properties of the specimens are presented in Table 3.

It was clearly seen that the greatest enhancement in the tensile strength and energy absorption can be noticed at the nanocomposite with MWCNT content of 0.25 wt%. With an addition of 0.25 wt% MWCNTs, the tensile strength increased from 428 ± 40.6 MPa to 507 ± 31.4 MPa corresponding to 180% enhancement; the energy at break increased from 20.8 ± 2.8 to 28.1 ± 0.8 J. This reinforcement effect is due to the relatively good interfacial interaction between the matrix and MWCNTs leading to improvement of load transfer [14,27,28]. Such an effect was also observed in nanocomposites with epoxy matrix filled with CNTs [29]. However, at higher content than 0.25 wt% of MWCNTs, the tensile strength and energy at break tended to decrease, which is probably due to agglomeration of the filler at this higher content, resulting in weak matrix-filler interfacial interaction [25]. The highest tensile strength obtained from this nanocomposite laminate filled with 0.25 wt% MWCNTs was higher than those of other composites reported in literature, for example, epoxy/Kevlar fiber filled with 0.5 wt% long length MWCNTs having 351 MPa [30] and epoxy/Kevlar filled with 0.32 wt% MWCNTs having 421 MPa [31]. Results shown in Table 3 reveal that the addition of MWCNTs has no significant improvement in tensile modulus, while a decrease in strain at break of the nanocomposite was observed, indicating the brittle characteristic of the specimen with MWCNTs. Okhawilai et al. reported that the nanocomposite with higher tensile strength resulted in higher ballistic impact resistance [8]. Consequently, the nanocomposite laminate containing 0.25 wt% MWCNTs was chosen to fabricate ballistic laminates for a further ballistic impact test and simulation study.

### 3.2. Surface Analysis by SEM of Poly(BA-a-co-PU) Matrix Reinforced with Aramid Fabric and MWCNTs

The morphological characteristics of the nanocomposite specimens with various MWCNT mass contents were studied by SEM. Figure 4a shows the pristine aramid fabric with appearance of gap between fibrils. After impregnated and compressed, the gap was completely filled by the matrix resin as illustrated in Figure 4b. The composite containing 0.25 wt% MWCNTs in the matrix poly(BA-a-co-PU) reinforced with aramid fabric showed smooth surface and well dispersion of MWCNTs as displayed in Figure 4c,d, respectively, at magnifications of 5000× and 30,000×. As MWCNT content exceeded 0.25 wt%, a rough surface and some agglomeration of MWCNTs were noticed, as seen in Figure 4e,f leading to a decrease in tensile strength, as a result of a reduction in matrix–filler interaction.

### 3.3. Ballistic Impact Tests on Nanocomposite Laminates

Ballistic tests were performed on laminates prepared with 0.25 wt% and 0.5 wt% of MWCNTs which showed the best tensile properties among the others and thus were chosen for further study of their ballistic impact performance. The ballistic impact tests on the nanocomposite laminates were experimentally carried out following NIJ-STD-0106.01 standard. One shot was conducted with 9 mm FMJ with an impact velocity of 394–408 m s^−1^. The ballistic impact resistances of the nanocomposite laminates with various numbers of plies, 10, 15 and 20, after impact are shown in Appendix A. If partial penetration occurred, the BFD of the nanocomposite laminate was then measured. Lower values of BFD imply better ballistic impact performance.

As shown in Table 4, it is clearly seen that 10-ply laminate without MWCNTs could not resist the complete penetration of the projectile, even at low velocity (333 m s^−1^) lower than the standard which is in the range of 394–407 m s^−1^. No perforation was observed in the nanocomposite laminates filled with MWCNTs, indicating that the inherent properties of MWCNTs enhance ballistic impact performance. Noticeably, the nanocomposite laminates filled with 0.25 wt% MWCNTs showed a higher ballistic impact performance than that filled with 0.5 wt% MWCNTs for all number of aramid fabric plies as revealed from the lower BFD. Evidently, 10 plies of aramid fabric with 0.25 wt% MWCNTs was able to stop the penetration of the 9 mm FMJ bullet at higher velocity of 407 m s^−1^, while the complete penetration or perforation was observed in the nanocomposite plate having 0.5 wt% MWCNTs even at lower velocity of 397 m s^−1^. At 15 and 20 aramid fabric plies, with 0.25 wt% and 0.5 wt% of MWCNTs, no perforation was noticed. These results show that the above laminates could resist the penetration from 9 mm FMJ bullet level II based on the NIJ-STD-0106.01 standard. In addition, the values of BFD were lower than 25 mm, which is considered as the maximum BFD established in the standard [20]. The nanocomposite laminate with 0.25 wt% MWCNTs, and 15 and 20 plies showed BFD of 10 and 8 mm. Moreover, it was found that the nanocomposite laminate with 0.25 wt% MWCNT has lower BFD than the nanocomposite with 0.5 wt%. For example, in the case of 15 plies, the BFD values were 10 and 12 mm for the composites with 0.25 wt% and 0.5 wt% MWCNTs, respectively. This result can be attributed to a better matrix–filler interfacial interaction of the nanocomposite laminate filled with 0.25 wt% MWCNTs resulting in enhanced energy absorption. These ballistic results are consistent with the higher tensile strength and energy absorption of the nanocomposite with 0.25 wt% than those with 0.5 wt%. The nanocomposite laminate with 20 plies of aramid fabric and 0.25 wt% MWCNTs nanocomposite has an areal weight density of 0.44 g cm^−2^ and thickness of 6.94 mm. These results identify this as the most suitable nanocomposite with high strength, energy absorption and low BFD, which can be further developed as a ballistic helmet.

### 3.4. Modeling of the Composite Panels

The nanocomposite laminate with 20 plies of aramid fabric impregnated with poly(BA-a-co-PU) 80/20 matrix containing 0.25 wt% MWCNTs was used for validation of the accuracy of parameters in the material model through the comparison between simulation and experimental results. The simulation test was carried out using the same conditions to the experimental work for better accuracy. The parameters of the material model are listed in Table 5, which were modified from ANSYS AUTODYN library. The material models were validated in the same manner as the experimental work using 9 mm FMJ bullet with velocity of 394 m s^−1^. The Mie–Gruneisen EOS was used for the material model of 9 mm FMJ bullet which was obtained from the material library of AUTODYN. In this experiment, the homogeneity properties in x and y directions were assumed to be equal. The failure was initially caused by excessive tensile stress or/and strain.

Both the experimental and simulation results showed that the bullet could not penetrate the nanocomposite laminate. The damage area, the depth of deformation and diameter of deformed bullet values obtained from the simulation were similar to experimental results as shown in Table 6, indicating the acceptable values of input parameters during simulation work. Figure 5 shows the comparison of the nanocomposite laminate and deformed bullet that occurred in the experimental and simulation tests. The errors of simulation to experimental results for damage area, depth of deformation of plate and deformed bullet diameter were 7.1%, 9.9%, and 13.7%, respectively.

### 3.5. BFD Prediction of Ballistic Helmet

The input parameters of laminates with 0.25 wt% MWCNTs obtained from Section 3.4 were used for simulation of ballistic helmet. According to the MIL-STD-662F, four impact locations, i.e., top, front, back, and side of the helmet, were impacted.

Figure 6 shows the deformation at the interior surface of the ballistic helmet after impact with the 9 mm FMJ bullet with a velocity of 358 m s^−1^. As shown in Figure 6, the helmet at all four locations could prevent the penetration of the projectile. The maximum BFD values at the top, front, rear and side of the helmet were found to be 8.19, 7.99, 7.21 and 6.94 mm, respectively. These deformation values are lower than those defined by the standard specification for a ballistic helmet, which are 16 mm for the front and rear areas of the helmet and 25 mm for the sides and top of the helmet—which are the standard specification values for ballistic helmets [20].

The plot of the inside surface deformation with time was presented in Figure 7 comparing the different impact locations on the helmet.

It can be seen from the slope of the graph that the top and front of the helmet were deformed faster than the side and rear zones. A comparison of BFD values obtained in this research with other ballistic helmet reports is shown in Table 7. Notably, our nanocomposite laminate had a lower thickness which is known to influence the ballistic impact performance, especially BFD. The values of BFD obtained from this research were lower than those reported by other researches using a simulation method indicating the potential use of aramid fabric impregnated with poly(BA-a-co-PU) 80/20 matrix containing 0.25 wt% MWCNTs for effective ballistic helmets [32,33].

### 3.6. Ballistic Limit Prediction of Ballistic Helmet

The ballistic limit (V_50_) is determined from the average of six projectile impact velocities, including three high velocities that result in partial penetration and three low velocities that cause complete penetration. The results of the ballistic helmet subjected to impacts with six different velocities of the bullet on the top of the helmet are presented in Table 8. The ballistic limit is determined to be at 632 m s^−1^. When the helmet was subjected to projectile with impact velocity lower than the ballistic limit, no perforation occurred as noticed in Figure 8a, while full penetration was observed when the impact velocity was higher than its ballistic limit, as seen in Figure 8b.

## 4. Conclusions

In this research, a lightweight nanocomposite ballistic helmet was developed based on aramid fabric impregnated with poly(BA-a-co-PU) 80/20 matrix containing 0.25 wt% MWCNTs having a protection level II following the NIJ-STD-0106.01 standard for ballistic helmets. The effect of MWCNT content on mechanical properties was investigated. It was found that tensile strength and energy at break of the nanocomposite with 0.25 wt% MWCNTs showed the highest values among the others. The ballistic impact performance of the nanocomposite laminates with different MWCNT contents and numbers of plies was also studied. The results confirmed the greatest ballistic impact resistance of the nanocomposite laminate filled with 0.25 wt% MWCNTs. Moreover, the BFD of the nanocomposite was lower with increasing number of plies. Interestingly, the 20-ply ballistic panels could withstand penetration of 9 mm FMJ and showed a BFD value of only 8 mm, which was much lower than 25 mm specified by NIJ-STD-0106.01. Furthermore, simulation models for ballistic impact on plate and helmet were studied using ANSYS AUTODYN. The comparison between experimental and simulation results confirmed the accuracy of the input parameters. The simulation results revealed that the nanocomposite helmet could resist the penetration of the 9 mm FMJ bullet at a velocity of 358 m s^−1^; also, low deformation at four impact locations, i.e., top, front, back and side of the ballistic helmet, was observed. The ballistic limit of the nanocomposite helmet was predicted to be 632 m s^−1^ according to MIL-STD-662F using FSP. The results indicated that the aramid fabric impregnated with poly(BA-a-co-PU) 80/20 matrix containing 0.25 wt% MWCNT nanocomposite developed in this study has great potential to be applied as a lightweight ballistic helmet with a protection level II of NIJ-STD 0106.01.

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
