# Peer review of "Development of Lightweight and High-Performance Ballistic Helmet Based on Poly(Benzoxazine-co-Urethane) Matrix Reinforced with Aramid Fabric and Multi-Walled Carbon Nanotubes"

_polymers, 2020, doi:10.3390/polym12122897_

Round 1
Reviewer 1 Report
The manuscript entitled “Development of Light Weight and High-performance Ballistic Helmet Based on Poly(benzoxazine-co-urethane) Matrix Reinforced with Aramid Fabric and Multi-walled Carbon Nanotubes” investigates the tensile and impact characteristics of laminated matrix composite reinforced with carbon nanotubes. The simulation was also carried out and the impact results were compared with those acquired from experimental. The idea is interesting however the manuscript needs a significant revision in term of writing.
Comments:
- The major limitation of this manuscript is the numerous grammatical and language errors, which need to be addressed. This significantly affected the quality of the manuscript and made it difficult to be followed. The authors are suggested to proofread the entire manuscript, to improve the language and clarity of it.
- Line 34, please remove one of “;” after Poly(benzoxazine-co-urethane)
- The authors should be careful about using abbreviations. The abbreviations must be defined in the first place. Some of them were defined several times.
- The authors only carried out the tensile test as one of the mechanical evaluations. The reviewer suggests the flexural tests will be also performed, otherwise, the “mechanical” term should be changed to tensile characteristics.
Author Response
Answer to reviewers
Reviewer: 1
The manuscript entitled “Development of Light Weight and High-performance Ballistic Helmet Based on Poly(benzoxazine-co-urethane) Matrix Reinforced with Aramid Fabric and Multi-walled Carbon Nanotubes” investigates the tensile and impact characteristics of laminated matrix composite reinforced with carbon nanotubes. The simulation was also carried out and the impact results were compared with those acquired from experimental. The idea is interesting however the manuscript needs a significant revision in term of writing.
Comments:
1) The major limitation of this manuscript is the numerous grammatical and language errors, which need to be addressed. This significantly affected the quality of the manuscript and made it difficult to be followed. The authors are suggested to proofread the entire manuscript, to improve the language and clarity of it.
Answer: The manuscript has grammatically corrected by Native English Professor.
2) Line 34, please remove one of “;” after Poly(benzoxazine-co-urethane)
Answer: Correct as suggested.
3) The authors should be careful about using abbreviations. The abbreviations must be defined in the first place. Some of them were defined several times.
Answer: The authors have checked abbreviations and corrected according to reviewer’s comment.
4) The authors only carried out the tensile test as one of the mechanical evaluations. The reviewer suggests the flexural tests will be also performed, otherwise, the “mechanical” term should be changed to tensile characteristics.
Answer: The authors have changed the word “mechanical” to “tensile” according to reviewer’s suggestion.

Reviewer 2 Report
This quite technical contribution has no major flaws. I do suggest to put the current work in a broader context on materials than can resist impact. This will enlarge the readability. Below some minor comments
L 40 ,.
L 57 please add the general reaction scheme
L 111 better average molar mass
L 115 alloy reads strange
L 119 so true copolymer or blend? Please highlight the chemical reactions.
L 173 why this Johnson-Cook model?
Author Response
Answer to Reviewer 2
Comments to the Authors:
This quite technical contribution has no major flaws. I do suggest to put the current work in a broader context on materials than can resist impact. This will enlarge the readability. Below some minor comments
1) L 40 ,.
Answer: Correct as suggested
2) L 57 please add the general reaction scheme
Answer: The general benzoxazine synthesis scheme is added as seen in Scheme 1.
3) L 111 better average molar mass
Answer: Correct as suggested
4) L 115 alloy reads strange
Answer: The word “alloy” has changed to “compound”.
5) L 119 so true copolymer or blend? Please highlight the chemical reactions.
Answer: The chemical reaction of benzoxazine resin and urethane prepolymer is shown in scheme 2.
6) L 173 why this Johnson-Cook model?
Answer: For V50 determination, the fragment simulating projectile (FSP) was used which was made from 4330 steel. The behavior of steel under ballistic impact follows Johnson-Cook model describing large strains, high strain rates and high temperature. The Johnson Cook model has widely used for FSP simulation as follows:
References
- James O Daniel, Nicholas Boone, Modeling fragment simulating projectile penetration into steel plates using finite elements and meshfree particles. Shock and Vibration 2011. 18: p. 425-436.
- Aare, M. and S. Kleiven, Evaluation of head response to ballistic helmet impacts using the finite element method. International Journal of Impact Engineering, 2007. 34(3): p. 596-608.

Round 2
Reviewer 1 Report
The manuscript can be recommended for publication in Polymers at the present form.